# Vitamin E Levels in Ethnic Communities in Malaysia and Its Relation to Glucose Tolerance, Insulin Resistance and Advanced Glycation End Products: A Cross-Sectional Study

**DOI:** 10.3390/nu12123659

**Published:** 2020-11-27

**Authors:** Geoffrey Hong Iing Chua, Sonia Chew Wen Phang, Yin Onn Wong, Loon Shin Ho, Uma Devi Palanisamy, Khalid Abdul Kadir

**Affiliations:** Jeffrey Cheah School of Medicine and Health Sciences, Monash University Malaysia, Petaling Jaya 47500, Selangor, Malaysia; sonia.phang1@monash.edu (S.C.W.P.); yin.wong@monash.edu (Y.O.W.); ho.loon.shin@monash.edu (L.S.H.); umadevi.palanisamy@monash.edu (U.D.P.)

**Keywords:** vitamin E, tocopherol, insulin resistance, β-cell function, receptor of advanced glycation end-products (RAGE), pre-diabetes, type 2 diabetes, Orang Asli (OA)

## Abstract

Malaysian national morbidity surveys on diabetic prevalence have shown ethnical variation among prediabetic and diabetic populations. In our attempt to understand this variation, we studied the α-tocopherol, insulin resistance, β-cell function and receptor of advanced glycation end-products (RAGE) levels, as risk factors of type 2 diabetes, among the different ethnicities. In total, 299 subjects of Malay, Chinese, Indian and aboriginal Orang Asli (OA) heritage were recruited from urban and rural areas of Malaysia by stratified random sampling. Serum α-tocopherol concentrations were measured using high performance liquid chromatography (HPLC) and insulin concentrations were measured using enzyme-linked immunosorbent assay (ELISA). In subjects with pre-diabetes, OAs had the highest α-tocopherol level, followed by Chinese and Malays (0.8938, 0.8564 and 0.6948 respectively; *p* < 0.05). In diabetic subjects, Malays had significantly higher RAGE levels compared to Chinese and Indians (5579.31, 3473.40 and 3279.52 pg/mL respectively, *p* = 0.001). Low α-tocopherol level (OR = 3.021, *p* < 0.05) and high insulin resistance (OR = 2.423, *p* < 0.05) were linked strongly to the development of pre-diabetes. Low β-cell function (OR = 5.657, *p* < 0.001) and high RAGE level (OR = 3.244, *p* < 0.05) were linked strongly to the development of diabetes from pre-diabetes. These factors might be involved in the development of diabetes, along with genetic and environmental factors.

## 1. Introduction

Type 2 diabetes has become a major public health problem globally. According to the World Health Organization (WHO, 20140), approximately 422 million adults aged 18 years and older worldwide have diabetes, and the global prevalence has almost doubled from 4.7% in 1980 to 8.5% in 2014 [1]. The rapid increase in the prevalence of diabetes worldwide is concerning as diabetes is associated with debilitating complications. Diabetes is the leading cause of cardiovascular diseases, blindness, chronic renal failure and lower limb amputation in many countries [2].

The prevalence of diabetes is also increasing rapidly. The Malaysia National Health and Morbidity Survey (NHMS) 2015 showed the prevalence of diabetes increased from 11.6% in 2006 to 17.5% in 2015 and to 18.3% in 2019 [3,4]. The prevalence of impaired fasting glucose or pre-diabetes in Malaysia remained stable at 4.2%, 4.9% and 4.7% in 2006, 2011 and 2015, respectively.

In Malaysia, Indians have the highest diabetes prevalence (22.1%), followed by Malays (14.6%), Chinese (12%) and indigenous people (2%) [3]. In Singapore, a neighboring country across the Johor Strait, a similar trend is seen whereby Indians have the highest prevalence (17.2%), trailed by Malays (16.6%) and Chinese (9.7%) [5]. This ethnical breakdown of diabetes prevalence illustrates the difference in risk of having diabetes in different ethnicities, despite living in a similar environment. In a study carried out by our group in Johor, Malays did not show correlation between the prevalence of diabetes and dietary factors, specifically daily total calorie intake and daily intake of carbohydrates, fats and proteins, as well as micronutrient intake (retinol, thiamine) [6]. However, there were small correlations with daily physical activity, socioeconomic groups and educational levels. As such, the ethnical variation in prevalence rates of diabetes and dysglycemia could be due to genetically conferred protection and/or other factors, including gut microflora and its dysbiosis, untested nutrients such as chromium, vitamin A and vitamin E, as well as inflammatory markers such as advanced glycation end products (AGE).

The indigenous people of Malaysia, the Orang Asli (OA “original people”), comprise only 0.76% of the population in Peninsular Malaysia [7]. While diseases such as malaria, tuberculosis, typhus, parasitic infections, iodine deficiency-induced goiter and malnutrition used to be endemic in the 1970’s and 1980’s, the rapid modernization of the country has led to socio-economic changes in certain OA populations. Consequently, epidemiological transition has been correlated with growing rates of non-communicable diseases. This was evident in a recent study by Aghakhanian et al. in 2018. They showed that the prevalence of metabolic syndrome (MetS) was highest in the Proto-Malays (39.56%) who live in settlements and semi urban areas, compared to the Negritos (26.35%) and the Senois (11.26%), who live in the jungles and are hunter gatherers [8].

Type 2 diabetes is a progressive endocrine disease caused by a combination of failure of the β-cells of the pancreas to secrete enough insulin and the failure of the body to utilize insulin effectively [9]. In response to insulin-resistant states, such as puberty, pregnancy and obesity, healthy human pancreatic β-cells can secrete up to four to five times more insulin, which maintains a normal glucose level [10,11,12]. However, with increasing insulin resistance and decreasing β-cell function, there is decreased insulin-dependent glucose uptake by the skeletal muscles and the liver, as well as ineffective suppression of hepatic glucose production [13,14]. This leads to hyperglycemia, impaired glucose tolerance and, eventually, type 2 diabetes. Both genetic and environmental factors interact in the development of diabetes, as genetic predisposition increases the susceptibility of insulin sensitivity and pancreatic β-cells towards detrimental environmental factors, such as obesity [15]. Obesity results in increased inflammatory and oxidative states, which in turn cause increased insulin resistance and decreased β-cell function, leading to the development of diabetes.

Recent studies have shown the receptor for advanced glycosylation end products (RAGE) to be directly involved in the pathogenesis of diabetes. Lee et al. found that RAGE and its ligands induced the apoptotic death of human islet cells and INS-1 cells (rat pancreatic β-cell line) [16]. RAGE knockdown is an experimental technique that inhibits the expression of RAGE genes. The study by Zhu et al. found that RAGE ligand-glycated serum induced the upregulation of RAGE in INS-1 cells and induced apoptosis in the β-cell line [17]. Both antibodies of RAGE and RAGE knockdown inhibited these β-cell toxic effects of glycated serum, hence directly involving RAGE in β-cell death. These findings not only apply to type 1 diabetes, but also to type 2 diabetes, as β-cell dysfunction is one of the hallmarks of the pathogenesis of type 2 diabetes.

Vitamin E occurs naturally as tocopherols and tocotrienols, each with α-, β-, γ- and δ- isoforms [18]. Studies have shown a correlation of lower plasma α-tocopherol levels with a higher prevalence of type 2 diabetes. A 4-year cohort study in Eastern Finland has highlighted that with every 1 μmol/L decrease in plasma α-tocopherol concentration, there is a 22% increased diabetes risk (*p* = 0.0004) [19]. The Insulin Resistance and Atherosclerosis Study (IRAS) showed similar findings in vitamin E supplement non-users, with an odds ratio for development of type 2 diabetes (for the highest quartile versus lowest quartile of α-tocopherol concentration) of 0.12 (95% CI 0.02–0.68, *p* < 0.01) [20]. Several studies have shown vitamin E tocopherols to increase insulin sensitivity by upregulating peroxisome proliferator-activated receptor (PPAR) activity [21,22], which is correlated with insulin resistance and type 2 diabetes when its activity is disrupted [23].

This study aims to determine the α-tocopherol, insulin resistance, β-cell function and RAGE levels, as risk factors of type 2 diabetes, and compare these levels among the Malay, Chinese, Indian and OA subjects, associating it with the stage of diabetic development, i.e., non-diabetes, pre-diabetes and diabetes.

## 2. Materials and Methods 

### 2.1. Study Design

This is a cross-sectional study of 299 subjects recruited in 2018/2019. The sample size calculation was based on a two-sided confidence interval of 95%, power of 80% and ratio of sample size of 1:1 to detect a mean difference of serum-corrected α-tocopherol concentration between non-diabetic (μ: 1.005; σ: 0.206) and diabetic subjects (μ: 0.906; σ: 0.178), as seen in the Finnish Study [19]. To fulfill these specifications, 60 subjects were required in each of the non-diabetes and diabetes group. The sample size calculations were done using the Open EPI Toolkit available at https://www.openepi.com/Menu/OE_Menu.htm [24].

Non-diabetic, pre-diabetic and diabetic subjects were recruited from community health screening programs conducted in the Johor Bahru, Masai and Kulai districts of Johor, Malaysia, as well as from the Clinical Research Centers of Jeffrey Cheah School of Medicine and Health Sciences, Monash University Malaysia in Bandar Sunway, Selangor state. The definition of type 2 diabetes status was based on the Australian Type 2 Diabetes Guidelines [25,26]—the non-diabetes group having HbA1c ≤ 5.9%, the pre-diabetes group having HbA1c between 6.0% and 6.49%, and the diabetes group having HbA1c ≥ 6.5%. 

The inclusion criteria were as follows: (1) age > 18 years old; (2) consent given; (3) more than 500μL of stored sera being available; (4) data on demographics, anthropometric measurements, HbA1c, fasting blood glucose and lipid profile being available; (5) not on vitamin E supplements for the past 6 months; (6) not having any microvascular or macrovascular diabetic complications.

Within the non-diabetic, pre-diabetic and diabetic groups, the subjects were subdivided into subgroups of comparable numbers of subjects based on their ethnicities, as follows: Malay, Chinese, Indian and OA. 

The study was conducted in accordance with the Declaration of Helsinki, with extended approval obtained from Monash University Human Research Ethics Committee (Project number: 18149). Informed consent was obtained during community health screening clinics.

### 2.2. Laboratory Measurements of Serum α-Tocopherol Concentration

#### 2.2.1. Extraction of α-Tocopherol from Serum Samples

Sample sera were stored in identity marked Eppendorf tubes at −80 °C for ~2 years, and the α-tocopherol samples were stable kept at −80 °C for ~7 years [20].

The extraction of analytes from serum samples was published previously [21]. In total, 100 μL of 0.0625% ethanol-BHT was added to 200 μL of serum in an amber microfuge tube for deproteinization. The mixture was then vortexed for 15 s using Stuart SA8 Vortex Mixer (Cole-Palmer Ltd., Staffordshire, UK). Then, 1 mL of 0.005% n-hexane-BHT was added to the mixture and then vortexed using a D1008 Microcentrifuge (DLAB Scientific Inc., Riverside, CA, USA) and shaken alternately for 5 min to extract lipophilic analytes. The mixture was then centrifuged at 3000 g for 3 min using a Beckman Coulter Microfuge 16 Centrifuge (Beckman Coulter, Pasadena, CA, USA). An aliquot of 800 μL of supernatant was pipetted into an amber glass vial. The extraction was repeated twice with 0.005% ethanol–BHT. The purified extract was evaporated using Micro-Cenvac NB-503 IR Concentrator (N-Biotek, Bucheon-Si, Gyeonggi-do, South Korea) and then reconstituted with 100 μL 0.0625% ethanol–BHT by vortexing for 3 min. The extracts were then placed in the high-performance liquid chromatography (HPLC) autosampler compartment at 4 °C before analysis. The whole procedure was performed under low ambient light to minimize the degradation of the analyte.

#### 2.2.2. Measurement of Serum α-Tocopherol Concentration Using High Performance Liquid Chromatography (HPLC)

HPLC was used to quantify the α-tocopherol concentration of samples. Purified sera extract was injected onto a Poroshell 120 EC-C18 2.1 × 100 mm column, with a 2.7 μm particle size. The mobile phase containing a mixture of acetonitrile, tetrahydrofuran, methanol and 1% ammonium acetate solution at 68:22:7:3 was eluted at a flow rate of 0.3 mL/min, using the Agilent 1200 Series Reverse Phase HPLC System (Agilent Technologies, Santa Clara, CA, USA) containing a G1312A binary pump in isocratic mode, a G1365C MWD detector (at 300 nm), and a programmable G1367D HiP sampler with a temperature-controlled sample tray. Agilent Chemstation Version B.04.03-SP2 was used for peak integration, sample calibration and data analysis. For the analysis of serum α-tocopherol concentration, a 6-level calibration curve was constructed using α-tocopherol standards of concentrations ranging from 15 μmol/L to 90 μmol/L. An external standard with a known concentration of α-tocopherol was included in all runs. The sera extract was injected alternately with a mobile phase-containing blank sample. The degree of analytical background response of α-tocopherol in serum (minimal detection limit) was assessed by calculating the mean and standard deviations of the response of 212 blank samples; the detection limit was 2.113895 μmol/L (σ: 1.644104).

### 2.3. Laboratory Measurements of Serum Insulin Resistance, β-Cell Function and RAGE Concentration

Enzyme-linked immunosorbent assay (ELISA) kits were used to measure serum insulin levels (Elabscience E-EL-H2665, Houston, TX, USA) and serum RAGE levels (Elabscience E-EL-H0295, Houston, TX, USA) based on the manufacturer’s protocol. Measurements were done in duplicates and the optical density was measured using a BioTek EON High Performance Microplate Spectrophotometer (BioTek US, Winooski, VT, USA) at 450 nm. The intraassay and interassay coefficients of variation were 5.0% and 4.5%, respectively, for the human insulin kits, and were 5.0% and 5.8%, respectively, for the human RAGE kits.

The fasting serum insulin and blood glucose levels were used in the updated homeostatic model assessment (HOMA2) calculator, to compute the insulin resistance (HOMA2 IR) and β-cell function (HOMA2%B) [27].

### 2.4. Statistical Analysis

Statistical Package for Social Sciences (SPSS) version 25 (IBM SPSS Inc., Chicago, IL, USA) was used to conduct statistical analysis of data. 

As lipophilic molecules such as vitamin E are transported intravascularly along with lipids in the form of lipoproteins, serum lipid levels can affect serum α-tocopherol concentration [28]. Therefore, the corrected α-tocopherol concentration was used in the data analysis to exclude the effects of serum lipids on α-tocopherol. The correction method used was adapted from previous studies [20,29]. A regression-based equation was used to adjust the α-tocopherol levels for serum total cholesterol, and triglyceride was derived from the Jordan et al. study [29]:

Y_adj_ = y + B_1_(x_1_ − x_1,0_) + B_2_(x_2_ − x_2,0_)

Y_adj_: lipid-adjusted ɑ-tocopherol level; y: measured ɑ-tocopherol level; B_1_: regression coefficient for total cholesterol; x_1_: measured total cholesterol; x_1,0_: standard value of total cholesterol set at 5.18 mmol/L; B_2_: regression coefficient for triglyceride; x_2_: measured triglyceride; x_2,0_: standard value of triglyceride set at 1.25 mmol/L. The standard values of total cholesterol and triglyceride were chosen respectively as values that correspond to a low cardiovascular risk in the study population, as stated in the Jordan et al. study [29]. The lipid-corrected α-tocopherol concentration was then calculated as the ratio of the measured α-tocopherol concentration to an expected α-tocopherol concentration.

The data were tested for normality. A non-parametric Kruskal–Wallis test was used to compare the levels of serum α-tocopherol, HOMA2 IR, HOMA2%B and RAGE between the following: (i) non-diabetic, pre-diabetic and diabetic study subjects; (ii) non-diabetic, pre-diabetic and diabetic subjects in subgroups of Malay, Chinese, Indian and OA ethnicity; (iii) Malay, Chinese, Indian and OA subjects in subgroups of non-diabetes, pre-diabetes and diabetes. The data were presented in medians with interquartile range (IQR). Post hoc Dunn’s test was then used to pinpoint specific medians that showed significant difference from the others. Binary logistic regression was used to assess the association of corrected serum α-tocopherol level, HOMA2 IR, HOMA2%B, and serum RAGE concentration with the development of diabetes. The lipid-corrected α-tocopherol concentration, HOMA2 IR, HOMA2%B and serum RAGE concentration were dichotomized using the median as a cut off to maximize statistical power. Additionally, Spearman’s correlation was used to examine the correlation of corrected serum α-tocopherol concentration with HOMA2 IR, HOMA2%B and serum RAGE concentration.

## 3. Results

### 3.1. Comparisons of Biomarkers in the Overall Study Population

The median serum α-tocopherol and corrected serum α-tocopherol concentrations, HOMA2 IR, HOMA2%B and serum RAGE concentrations in the study population, grouped as non-diabetics, pre-diabetics and diabetics, are outlined in Table 1.

In the overall study population, there was no significant difference in serum α-tocopherol concentration between the population groups (*p* = 0.185). However, when serum α-tocopherol concentration was corrected based on lipid levels, a significant difference was observed (*p* = 0.008). Amongst the groups, non-diabetics had the highest corrected serum α-tocopherol concentration (0.9989; IQR: 0.3586). Pre-diabetics had a 21% lower corrected serum α-tocopherol concentration (0.7874; IQR: 0.2905) compared to the non-diabetics. Diabetics had a higher corrected serum α-tocopherol concentration (0.8585; IQR: 0.3537) than the pre-diabetics, albeit lower compared to the non-diabetics.

Significant difference was observed in insulin resistance (*p* = 0.007) and in β-cell function (*p* < 0.0000001) amongst the groups. Pre-diabetics (0.0217; IQR: 0.0355) had double the insulin resistance of non-diabetics (0.0112; IQR: 0.0148). Diabetics (0.0136; IQR: 0.0182) had a higher insulin resistance than the non-diabetics, but lower compared to pre-diabetics. β-cell function was the highest in pre-diabetic subjects (5.95; IQR: 8.28). This was followed by that of non-diabetic subjects (4.15; IQR: 5.35). β-cell function in diabetic subjects (1.90; IQR: 3.00) was the lowest amongst the groups, and was half that of the non-diabetic subjects and one-third that of the pre-diabetic subjects.

Serum RAGE concentration was the highest in diabetic subjects (4397.29 pg/mL; IQR: 3891.27) followed by pre-diabetic subjects (3502.14 pg/mL; IQR:501.44) and non-diabetic subjects (3445.68 pg/mL; IQR: 1262.43); the difference seen was statistically significant (*p* < 0.001).

### 3.2. Comparisons of Biomarkers in Subgroups

#### 3.2.1. Comparisons of Biomarkers in Malay, Chinese, Indian and OA Subjects in Subgroups of Non-Diabetics, Pre-Diabetic and Diabetics

Table 2 outlines the comparisons of uncorrected and corrected serum α-tocopherol, insulin resistance, β-cell function and serum RAGE concentration.

In the Malays, there was a significant difference in corrected serum α-tocopherol concentration across the diabetic stages (*p* = 0.003). The pattern was akin to that of the overall study population, where the corrected serum α-tocopherol concentration was the highest in non-diabetics (0.9698; IQR: 0.2129) and lowest in pre-diabetes (0.6948; IQR: 0.1702), and in diabetes (0.8542; IQR: 0.4077) the level was higher than in pre-diabetics but lower than in non-diabetics. There was also a significant difference in insulin resistance in the Malays across the diabetic stages (*p* = 0.038). The insulin resistance increased in an ascending order (non-diabetics (0.0014; IQR: 0.0056), pre-diabetics (0.0068; IQR: 0.0181) and diabetics (0.0100; IQR: 0.0149)), with the level being almost ten times higher in diabetics compared to non-diabetics. Across all diabetic stages, insulin resistance amongst the Malays was significantly lower compared to the overall study population (Table 1). A significant difference was also seen in serum RAGE concentration among the Malays across diabetic stages (*p* = 0.0003). The serum RAGE concentration in the diabetic Malays (5579.31 pg/mL; IQR: 3414.10) was significantly higher than in the non-diabetics (1284.58 pg/mL; IQR: 6195.65) and pre-diabetics (754.34; IQR: 3635.61). This was distinct compared to the overall population, where the RAGE levels were similar in non-diabetics and prediabetics and the increase in RAGE levels in overall diabetics was not as pronounced as that seen in the Malays (Table 1). 

In the Chinese population, which has the lowest prevalence of diabetes, a significant difference in insulin resistance was observed across the diabetic stages (*p* = 0.0004). This pattern was similar to the overall population where insulin resistance was highest among the pre-diabetics (0.0357; IQR: 0.0430). The main difference was that the insulin resistance was the lowest in diabetics (0.0109; IQR: 0.0156). In non-diabetics (0.0174; IQR: 0.0230), the level was higher than in the diabetics but lower than in the pre-diabetics. Notably, insulin resistance in the Chinese was significantly higher than that in the overall population in the non-diabetics and prediabetics groups, but not in the diabetic group. (Table 1). Additionally, a highly significant difference was observed in the β-cell function across the Chinese diabetic stages (*p* < 0.0000001). Corresponding to the insulin resistance, the β-cell function was also the highest in pre-diabetics (9.40, IQR: 7.05), followed by non-diabetics (8.55; IQR: 5.70) and diabetes (2.00; IQR: 2.60). The β-cell function of the Chinese was higher than the overall population across all diabetic stages (Table 1).

In the Indians, a significant difference was observed in the β-cell function in non-diabetics and diabetics (*p* = 0.008). In the present study, a comparable sample size could not be obtained for pre-diabetic Indians. The β-cell function of diabetic Indians (1.60; IQR: 3.33) was more than one-third that of the non-diabetic Indians (5.00; IQR: 1.50).

#### 3.2.2. Comparisons of Biomarkers in Non-Diabetic, Pre-Diabetic and Diabetic Subjects in Subgroups of Malay, Chinese, Indian and OA Ethnicit

Table 3 depicts the ethnic comparison of uncorrected and corrected serum α-tocopherol concentration, insulin resistance, β-cell function and serum RAGE concentration, grouped in non-diabetes, pre-diabetes and diabetes.

Significant difference was seen in the serum α-tocopherol concentration amongst the non-diabetic (*p* = 0.00003) and pre-diabetic (*p* = 0.023) Malay, Chinese, Indian and OA subjects. In non-diabetes, the Chinese (53.79; IQR: 45.76) had the highest serum α-tocopherol concentration, followed by the Malays (52.45; IQR: 38.31), the Indians (32.80; IQR: 20.40) and the OAs (25.54; IQR: 16.52). Notably, in non-diabetics, the serum α-tocopherol concentration in the Chinese was almost double that of the OAs. The trend was similar in pre-diabetes, where the Chinese (50.21; IQR: 37.11) had the highest serum α-tocopherol concentration, followed by the Malays (41.03; IQR: 36.99) and the OAs (26.80; IQR: 9.26). However, no significant difference was seen in the serum α-tocopherol concentrations of diabetic Malay, Chinese, Indian and OA subjects.

When the serum α-tocopherol concentration was lipid-corrected, a significant difference could only be seen in the pre-diabetic Malay, Chinese and OA subjects (*p* = 0.023). In pre-diabetes, the OAs (0.8938; IQR: 0.4498) had the highest corrected serum α-tocopherol concentration instead, followed by the Chinese (0.8564; IQR: 2440) and the Malays (0.6948; IQR: 0.1702). In pre-diabetes, the Malays had corrected serum α-tocopherol concentrations lower than the overall study population, while the Chinese and the OAs had levels higher than the overall study population. 

There was a significant difference in insulin resistance amongst non-diabetic (*p* = 0.003) and pre-diabetic (*p* = 0.005) Malay, Chinese, Indian and OA subjects. In non-diabetes, the Chinese (0.0174, IQR: 0.0233) had the highest insulin resistance, followed by the Indians (0.0113; IQR: 0.0056), the OAs (0.0083; IQR: 0.0119) and the Malays (0.0014; IQR: 0.0056). In non-diabetes, the insulin resistance of Chinese subjects was double that of the OAs and ten times that of the Malays. This trend continued in pre-diabetes, where the Chinese (0.0357; IQR: 0.0430) had the highest insulin resistance, followed by the OAs (0.0111; IQR: -) and the Malays (0.0068; IQR: 0.0181). In pre-diabetes, the insulin resistance of the Chinese was triple that of the OAs and five times that of the Malays. There was no significant difference in the insulin resistance of diabetic Malay, Chinese, Indian and OA subjects.

Significant differences can be seen in the β-cell function of Malay, Chinese, Indian and OA subjects in non-diabetes (*p* = 0.00006) and pre-diabetes (*p* = 0.0002). In non-diabetes, the Chinese (8.55; IQR: 5.70) had the highest β-cell function, followed by the Indians (5.00; IQR: 1.50), the OAs (2.20; IQR: 2.90) and the Malays (1.00; IQR: 3.10). In non-diabetes, the Chinese had β-cell functions four times that of the OAs and more than eight times that of the Malays. In pre-diabetes, the Chinese (9.40; IQR: 7.05) had the highest β-cell function, followed by the Malays (2.30; IQR: 2.30) and the OAs (2.10; IQR: -). In pre-diabetes, the Chinese had a β-cell function of more than four times that of the Malays and the OAs. There was no significant difference in β-cell function of diabetic Malay, Chinese, Indian and OA subjects.

There was no significant difference in the serum RAGE concentration in different ethnicities who were non-diabetic or pre-diabetic. However, a significant difference was seen in serum RAGE concentration in Malay, Chinese, Indian and OA subjects with diabetes (*p* = 0.001). In diabetic subjects, Malays (5579.31; IQR: 3414.10) had almost double the RAGE concentration compared to Chinese (3473.40; IQR: 3873.40) and Indian (3279.52; IQR: 1741.34). We could not obtain comprehensive measurement of serum RAGE concentration in the OA due to a technical problem with the ELISA kit.

Overall, in the non-diabetics, the Chinese had the highest uncorrected serum α-tocopherol concentration, insulin resistance and β-cell function. The Indians had the second highest uncorrected serum α-tocopherol concentration, insulin resistance and β-cell function. The Malays had the lowest insulin resistance and lowest β-cell function. The OAs had the lowest uncorrected serum α-tocopherol concentration. However, there were no significant differences found in corrected serum α-tocopherol concentration and serum RAGE concentration amongst the different ethnicities.

Among the pre-diabetics, the OAs had the highest lipid-corrected serum α-tocopherol concentration, even though they had the lowest uncorrected serum α-tocopherol concentration. Having the highest uncorrected serum α-tocopherol concentration, the Chinese only had the second highest corrected serum α-tocopherol concentration, which was slightly lower than the OAs. Again, in pre-diabetics, the Chinese had by far the highest insulin resistance and β-cell function. The OAs and the Malays had similarly low insulin resistance and β-cell function compared to the Chinese. There was no significant difference in serum RAGE concentrations amongst the pre-diabetic Malay, Chinese, Indian and OA subjects.

Finally, in the subjects with diabetes, the Malays had almost double the serum RAGE concentration of the Chinese and the Indians. There were no significant differences in uncorrected and corrected serum α-tocopherol concentration, insulin resistance and β-cell function amongst the diabetic subjects of Malay, Chinese, Indian and OA ethnicities.

### 3.3. Correlations of Biomarkers to Stages of Development of Diabetes

Table 4 highlights the correlation of corrected serum α-tocopherol concentration, insulin resistance, β-cell function and serum RAGE concentration to different stages of development of type 2 diabetes.

The low corrected serum α-tocopherol concentration was correlated strongly with pre-diabetics, compared to non-diabetics, with more than four times the likelihood (OR = 4.192; 95% CI: 2.005, 8.765; *p* < 0.001). High insulin resistance (OR = 2.423; 95% CI: 1.052, 5.582; *p* < 0.05) was correlated with a more than two times higher likelihood of having pre-diabetes compared to non-diabetes.

Low β-cell function was correlated strongly with diabetics compared to pre-diabetics, with more than five times the likelihood (OR = 5.657; 95% CI: 2.397, 13.349; *p* < 0.001). A high serum RAGE concentration was linked with a more than three times higher likelihood (OR = 3.244; 95% CI: 1.454, 7.236; *p* < 0.05) of having diabetes compared to pre-diabetes.

A low corrected serum α-tocopherol concentration was linked with almost three times the likelihood (OR = 2.885; 95% CI: 1.654, 5.031; *p* < 0.001) of diabetes compared to non-diabetes. High insulin resistance was correlated with almost two times the likelihood (OR = 1.903; 95% CI: 1.013, 3.573; *p* < 0.05) of diabetes compared to non-diabetes. Low β-cell function (OR = 3.046; 95% CI: 1.609, 5.766; *p* = 0.001) and high serum RAGE concentration (OR = 2.870; 95% CI: 1.498, 5.496; *p* = 0.001) were linked to diabetes, compared to non-diabetes, with three times higher likelihood.

### 3.4. Correlation of Corrected Serum α-Tocopherol Concentration with Other Biomarkers

Table 5 depicts the correlation of corrected serum α-tocopherol concentration with insulin resistance, β-cell function and serum RAGE concentration. The corrected serum α-tocopherol concentration did not have any significant correlation with insulin resistance (r_s_ = −0.029; *p* = 0.699), β-cell function (r_s_ = 0.077; *p* = 0.308) or serum RAGE concentration (r_s_ = 0.035; *p* = 0.633).

## 4. Discussion

In the present study, there was no significant difference in uncorrected serum α-tocopherol concentration between the non-diabetics, pre-diabetics and diabetics (*p* = 0.185). However, when the serum α-tocopherol concentration was corrected based on the corresponding lipid levels of the study subjects, a significant difference was seen (*p* = 0.000002). This might be because vitamin E is lipophilic and is transported in the blood with cholesterols and triglycerides [28]. The lipid-correction done on the serum α-tocopherol concentration excludes the effects of lipid levels on α-tocopherol concentration, thus making the corrected concentration a better representation of a subject’s α-tocopherol level. 

The lowest corrected serum α-tocopherol concentration was found in the pre-diabetic stage. In diabetes, the corrected serum α-tocopherol concentration was also lower than non-diabetes. This crucial information shows that a lower than normal (non-diabetes) corrected serum α-tocopherol concentration is implicated in the development of diabetes, especially in the stage of pre-diabetes. Moreover, the data from this study show that a low corrected serum α-tocopherol concentration was correlated with a more than four times likelihood of having pre-diabetes compared to non-diabetes (OR = 4.192; *p* < 0.001). Many previous studies, such as the Eastern Finland study [19] and IRAS [20], have shown low α-tocopherol concentration to be a risk factor for the development of type 2 diabetes. Although the current study also found that a low corrected serum α-tocopherol concentration was involved in the development of diabetes (OR = 2.885; *p* < 0.001), the data also show that it was specifically implicated in an earlier stage, in pre-diabetes. This contrasts with previous studies, which only correlated it with the development of frank diabetes. From these findings, we speculate that a low corrected serum α-tocopherol concentration is a risk factor in the development of pre-diabetes. A main difference between the study population of the Eastern Finnish study and the current study is that their study population had a significantly lower α-tocopherol concentration (mean: 19 μmol/L) compared to our study population (mean: 41 μmol/L). Although the mechanism is not known, it is possible that the difference in α-tocopherol concentration between the two populations leads to the differences seen in the risk profiles of development of type 2 diabetes.

Increasing insulin resistance and decreasing β-cell function are hallmark events that lead to the development of impaired glucose tolerance and, eventually, type 2 diabetes [14]. Data from the current study showed that insulin resistance was 120% higher (0.0136 vs. 0.0112; *p* < 0.05) in diabetic subjects in comparison with non-diabetic subjects. Additionally, the β-cell function in diabetic subjects was half that of non-diabetics (1.90 vs. 4.15; *p* < 0.001). Study subjects with high insulin resistance (OR = 1.903, *p* < 0.05) and low β-cell function (OR = 3.046, *p* = 0.001) also had a higher likelihood of being diabetic than being non-diabetic. These findings are in keeping with previous findings that a combination of high insulin resistance and low β-cell function are key factors in the development of type 2 diabetes.

A closer examination of the relationship of insulin resistance and β-cell dysfunction with the stages of development of type 2 diabetes yields valuable information. Pre-diabetic study subjects in the current study were characterized by having the highest insulin resistance (0.0217; *p* < 0.05) compared to the non-diabetics and diabetics. Inversely, β-cell function was at the highest in our pre-diabetic subjects (5.95; *p* < 0.001). This means that in our study population, β-cell compensation can still be seen in pre-diabetes. As suggested by a previous study, β-cell function has to be assessed in correlation with insulin resistance [30]. On the other hand, extrapolated data from the UKPDS suggest that β-cell dysfunction precedes the diagnosis of type 2 diabetes by up to 12 years [31]. In other words, the extrapolated data speculate that a decline in β-cell function is seen early in pre-diabetes or even in non-diabetes, as a meta-analysis in Australia found the average duration of pre-diabetes in males and females >30 years old to be 8.5 years and 10.3 years, respectively [32]. This is a stark contrast compared to the findings in the current study of an increased β-cell function to compensate for high insulin resistance in pre-diabetes. The extrapolation of data in the UKPDS could be debatable, as it might be imprecise and thus fail to capture the actual picture of the progression of β-cell function decline and diabetes. Further prospective or genetic studies need to be done to validate the exact timeline of β-cell function decline in the progression of diabetes from pre-diabetes.

The current study also looked at odds ratios of insulin resistance and β-cell function in non-diabetes, pre-diabetes and diabetes to quantify the risk factors. Data from the current study found that subjects with high insulin resistance (OR = 2.423; *p* < 0.05) were linked with a more than twofold likelihood of having pre-diabetes compared to non-diabetes. On the other hand, subjects with low β-cell function have six times higher likelihoods of having diabetes compared to pre-diabetes (OR = 5.657; *p* < 0.001). These findings lead us to postulate that in Asians, high insulin resistance is an important risk factor in the development of pre-diabetes from non-diabetes, while low β-cell function is more crucial in the development of diabetes from pre-diabetes. To support our hypothesis, a recent study in China similarly found that insulin resistance and β-cell dysfunction are important determinants in the development of pre-diabetes and diabetes, respectively [33].

Chronic hyperglycemia leads to the formation of AGE [34]. AGE then interacts with its signal-transduction receptor, and through a series of events, leads to macrovascular and microvascular diabetic complications [35]. Other than causing diabetic complications, several studies also found that RAGE may be directly implicated in the pathogenesis of diabetes through β-cell toxicity [16,17]. 

In the current study, there was a modest 2% higher median serum RAGE concentration in pre-diabetics compared to non-diabetics (3502.14 vs. 3445.68; *p* < 0.001). In contrast, diabetics had a 25% higher median serum RAGE concentration compared to pre-diabetics (4397.29 vs. 3502.14; *p* < 0.001). Data from the current study also found that subjects with high serum RAGE concentration were three times more likely to have diabetes compared to pre-diabetes (OR = 3.244; *p* < 0.05). The key question now is whether a high serum RAGE concentration is a risk factor for the development of diabetes from pre-diabetes due to RAGE’s effect on β-cell function, or whether a high serum RAGE concentration is a direct consequence of chronic hyperglycemia from the development of frank diabetes. Although we can see that a high serum RAGE concentration is linked to the development of diabetes from pre-diabetes, more studies need to be conducted to understand the mechanism behind this relationship.

In addition, the present study also found no significant correlation of insulin resistance, β-cell function and serum RAGE concentration with corrected serum α-tocopherol concentration. This leads us to ponder the exact mode of action of α-tocopherol in diabetes. More studies at a molecular level need to be done to find out the exact pathways of α-tocopherol that link with diabetes, such as studies that look at pro-inflammatory and PPAR transcription factors.

The current study provides valuable information on the ethnical variation in the risk profile of the development of pre-diabetes and diabetes in terms of the biomarkers investigated.

Based on the data of the current study, in pre-diabetes, the Malays (0.6948; *p* = 0.023) have the lowest corrected serum α-tocopherol concentration compared to the Chinese (0.8564; *p* = 0.023) and the OAs (0.8938; *p* = 0.023). The α-tocopherol level in pre-diabetic Malays is also 12% lower (0.6948 vs. 0.7874) than the overall population. This trait might pose an additional risk of diabetes, as having the lowest corrected serum α-tocopherol concentration is shown to be a risk factor of pre-diabetes and diabetes. The majority of the subjects in the present study did not take vitamin E supplements. As such, it is possible that this variation in α-tocopherol concentration could be due to differences in lifestyle and dietary intake, or due to genetic variation leading to differences in the metabolism of the antioxidant.

The Malays have significantly lower insulin resistance compared to the overall population across all diabetic stages. Correspondingly, the Malays’ β-cell function is significantly lower than other ethnicities’ in terms of non-diabetes (*p* < 0.001) and pre-diabetes (*p* < 0.001). Additionally, their insulin resistance increases steadily from non-diabetes to pre-diabetes and to diabetes, a stark difference compared to the trend seen in the overall population, where insulin resistance peaks in pre-diabetes. Considering high insulin resistance as a risk factor of pre-diabetes, this is a favorable trait for the Malays. 

On the other hand, although the trends of increase in serum RAGE concentration are similar between the Malays and the overall study population, the degree of increase in serum RAGE concentration between pre-diabetes and diabetes is much higher in Malays (~700%) compared to the study population (~125%). It is also important to note that there are no significant differences seen in serum RAGE concentration in other ethnicities as diabetes progresses from non-diabetes to pre-diabetes and eventually diabetes. The current study also finds a significant difference in serum RAGE concentration between ethnic groups who are diabetic (*p* = 0.001). In diabetes, the Malays (5579.31 pg/mL) have almost double the serum RAGE concentration of the Chinese (3473.40 pg/mL) and of the Indians (3279.52 pg/mL). Additionally, no significant difference is found in serum RAGE concentration across the diabetic stages in the Chinese and Indians. The RAGE concentration of the OAs could not be obtained due to a technical problem with the ELISA kit. Due to their high serum RAGE concentration, the Malays may have a higher risk of diabetes due to the β-cell toxicity of RAGE. These findings regarding RAGE levels may also suggest Malays could be at a higher risk of diabetic complications as the AGE–RAGE interaction can lead to endothelial dysfunction due to increased inflammation and oxidative stress.

In the present study, among the pre-diabetics, the Chinese had the second highest corrected serum α-tocopherol concentration (0.8564; *p* = 0.023) compared to the other ethnicities. The corrected serum α-tocopherol concentration of pre-diabetic Chinese was also 8% higher (0.8564 vs. 0.7874) than the overall study population. This higher α-tocopherol level could be a protective factor against pre-diabetes for the Chinese.

Data from the present study show that the Chinese had significantly higher insulin resistance compared to the other ethnicities in non-diabetics (0.0174; *p* = 0.003) and pre-diabetics (0.0357; *p* = 0.005). Compared to the overall study population, their insulin resistance was 55% higher in non-diabetics (0.0174 vs. 0.0112) and 65% higher (0.0357 vs. 0.0217) in pre-diabetics. As a result, the β-cell function of the Chinese is correspondingly high in non-diabetes (8.55; *p* < 0.001) and in pre-diabetes (9.40; *p* < 0.001), compared to the other ethnicities. As discovered in the current study, having higher insulin resistance puts the Chinese at a higher risk of the development of diabetes from the pre-diabetic stage.

In non-diabetics, the Indians in the current study had the second highest insulin resistance (0.0113; *p* = 0.003), and correspondingly, the second highest β-cell function (5.00; *p* < 0.001) compared to the other ethnicities. However, when compared to the overall study population, the data showed that the Indians’ insulin resistance (0.0113 vs. 0.0112) and β-cell function (5.00 vs. 4.15) were similar to the overall. This might indicate that the Indians have an average risk profile compared to the overall population in terms of insulin resistance and β-cell function. A comparable sample size could not be obtained for the pre-diabetic Indians.

In the current study, the pre-diabetic OAs had the highest corrected serum α-tocopherol concentration (0.8938; *p* = 0.023). The corrected serum α-tocopherol concentration of pre-diabetic OAs was 13% higher (0.8938 vs. 0.7874) compared to the overall study population. As such, like the Chinese, the higher α-tocopherol level could protect the pre-diabetic OAs against the progression to diabetes.

Data from the present study found that the OAs had lower insulin resistance in comparison with other ethnicities in non-diabetes (0.0083; *p* = 0.003) and pre-diabetes (0.0111; *p* = 0.005). Additionally, in comparison with the overall study population, the OAs’ insulin resistance was lower in non-diabetes (0.0083 vs. 0.0112) and pre-diabetes (0.0111 vs. 0.0217). Correspondingly, the OAs’ β-cell function was low in non-diabetes (2.20; *p* < 0.001) and pre-diabetes (2.10; *p* < 0.001), compared to the other ethnicities. There were also no significant differences in the OAs’ insulin resistance and β-cell function across diabetic stages, as they remained low. The low insulin resistance can be a protective trait against diabetes for the OAs.

These examples show the potential of an individualized therapy at different stages of diabetes based on the risk factor each person exhibits, when the biomarkers of the diabetes risk factors tested in this study or newer biomarkers become more accessible. Notably, in diabetes, the ethnical differences in corrected α-tocopherol concentration, insulin resistance and β-cell function seen in non-diabetes and pre-diabetes were not significant. Moreover, a significant difference in corrected α-tocopherol concentration amongst multiple ethnicities was seen only in pre-diabetes. As such, secondary prevention efforts for individuals with pre-diabetes could address this and close the ethnical gap in diabetic risk factors. In other words, this makes pre-diabetes a suitable phase for interventional efforts, such as vitamin E supplementation, to improve the α-tocopherol levels as well as to better the insulin resistance and β-cell function. In addition, the long duration of pre-diabetes of up to 8–10 years [32] provides ample opportunity to improve glucose tolerance. If the interventions are proven to be cost effective, this could also warrant a population-wide screening program to detect individuals with pre-diabetes.

Malaysia is a multiethnic country, as each ethnic group has its own unique culture and cuisine. The multiethnic people consume a vast variety of diets comprising different compositions of macronutrients and micronutrients, as well as antioxidants, such as natural sources of α-tocopherol. This environmental factor, unique to this country, could to some degree contribute to the difference in the risk profiles seen. Additionally, the disparity in access to healthcare facilities and in education level can give rise to different degrees of diabetic education and diabetic treatment. This could also bring about the difference in the diabetes risk factors studied, such as insulin resistance, β cell function and serum RAGE levels. Other than that, it is increasingly recognized that age, hyperglycemia [36] and obesity [37] are risk factors for the development of AGE and, in turn, RAGE. However, due to the limitation of the variation in the data collected, these factors could not be eliminated using parametric statistical tests.

All in all, the differences in corrected α-tocopherol concentration, insulin resistance, β-cell function and serum RAGE concentration in subjects of different ethnicities gives rise to different risk profiles of diabetes. As diabetes is a multifactorial disease, these factors contribute to different degrees to the differences in diabetes prevalence seen in national surveys, along with other genetic and lifestyle factors. 

## 5. Conclusions

The current study found that the OAs had the highest corrected serum α-tocopherol concentration in the pre-diabetics, followed by the Chinese and the Malays. The Chinese have the highest while the Malays have the lowest insulin resistance and β-cell function, compared to the other ethnicities in the non-diabetics and pre-diabetics. The diabetic Malays have significantly higher serum RAGE concentrations compared to the non-diabetic and pre-diabetic Malays, as well as in comparison with other ethnicities. The differential risk profiles seen in different ethnicities can give rise to the ethnical variation in diabetes prevalence seen in national surveys to different degrees along with other genetic and environmental risk factors, as diabetes is a multifactorial disease.

The ethnical variations in the corrected α-tocopherol concentration, insulin resistance and β-cell function seen in non-diabetic and pre-diabetic subjects were not significant in the diabetic subjects. This signifies that an interventional effort should be made in the pre-diabetic stage to close the ethnical gaps in risk factors, before it is too late.

We also postulate that a low corrected serum α-tocopherol concentration and a high insulin resistance are key factors in the development of pre-diabetes. Conversely, low β-cell function and high serum RAGE concentration are linked strongly to the development of diabetes. Prospective cohort studies should be done in the future to investigate causality more precisely.

## Figures and Tables

**Table 1 nutrients-12-03659-t001:** The median serum concentration of α-tocopherol (μmol/L), corrected serum concentration of α-tocopherol, HOMA2 IR, HOMA2%B and serum RAGE concentration (pg/mL) between non-diabetic, pre-diabetic and diabetic subjects.

	Non-Diabetes (*n* = 90)	Pre-Diabetes (*n* = 42)	Diabetes (*n* = 118)	*p* Value ^a^
Serum concentration of α-tocopherol (IQR) (μmol/L)	33.72 (31.01)	39.02 (35.84)	41.30 (36.65)	0.185
Corrected serum concentration of α-tocopherol (IQR)	0.9989 (0.3586)	0.7874 (0.2905)	0.8585 (0.3537)	0.000002 **
HOMA2 IR (IQR)	0.0112 (0.0148)	0.0217 (0.0355)	0.0136 (0.0182)	0.007 *
HOMA2%B (IQR)	4.15 (5.35)	5.95 (8.28)	1.90 (3.00)	0.00000008 **
Serum RAGE concentration (IQR) (pg/mL)	3445.68 (1262.43)	3502.14 (501.44)	4397.29 (3891.27)	0.0003 **

HOMA2 IR: Insulin resistance; HOMA2%B: β-cell function; RAGE: Receptor of advanced glycation end products. ^a^ Kruskal-Wallis test was applied. Assumptions were fulfilled. * Signifiant at *p* < 0.05, ** Significant at *p* < 0.001 Post-hoc Dunn test was applied. For corrected serum concentration of α-tocopherol, significant difference was found between pre-diabetes and non-diabetes (*p* < 0.0001) as well as between diabetes and non-diabetes (*p* = 0.001). For HOMA2 IR, significant difference was found between pre-diabetes and non-diabetes (*p* = 0.006). For HOMA2%B, significant difference was found between non-diabetes and diabetes (*p* < 0.0001) as well as between diabetes and pre-diabetes (*p* < 0.0001). For serum RAGE concentration, significant difference was found between non-diabetes and diabetes (*p* = 0.0003) as well as between pre-diabetes and diabetes (*p* = 0.033).

**Table 2 nutrients-12-03659-t002:** Comparison of the median serum concentration of α-tocopherol (μmol/L), corrected serum concentration of α-tocopherol, HOMA2 IR, HOMA2%B and serum RAGE concentration (pg/mL) in non-diabetes, pre-diabetes and diabetes, grouped in different ethnic groups.

	Malay				Chinese				Indian				OA			
	Non-Diabetes (*n* = 20)	Pre-Diabetes (*n* = 13)	Diabetes (*n* = 45)	*p* Value	Non-Diabetes (*n* = 20)	Pre-Diabetes (*n* = 21)	Diabetes (*n* = 35)	*p* Value	Non-Diabetes (*n* = 17)	Pre-Diabetes	Diabetes (*n* = 29)	*p* Value	Non-Diabetes (*n* = 52)	Pre-Diabetes (*n* = 11)	Diabetes (*n* = 5)	*p* Value
Serum α-tocopherol concentration ^a^ (IQR) (μmol/L)	52.45 (38.31)	41.03 (36.99)	42.01 (42.55)	0.602	53.79 (45.76)	50.21 (37.11)	50.17 (20.40)	0.772	32.80 (20.40)	NA ^c^	30.64 (29.30)	0.587	25.54 (16.52)	26.80 (9.26)	34.05 (17.90)	0.418
Corrected serum α-tocopherol concentration ^a^ (IQR)	0.9698 (0.2129)	0.6948 (0.1702)	0.8542 (0.4077)	0.003 *	0.9563 (0.2252)	0.8564 (0.2440)	0.8596 (0.3601)	0.058	0.9027 (0.4145)	NA ^c^	0.8496 (0.3837)	0.140	1.0614 (0.4805)	0.8938 (0.4498)	0.8911 (0.3764)	0.052
HOMA2 IR ^a^ (IQR)	0.0014 (0.0056)	0.0068 (0.0181)	0.0100 (0.0149)	0.038 *	0.0174 (0.0230)	0.0357 (0.0430)	0.0109 (0.0156)	0.0004 *	0.0113 (0.0056)	NA ^c^	0.0157 (0.0153)	0.553	0.0083 (0.0119)	0.0111 (-)	0.0308 (0.0302)	0.148
HOMA2%B ^a^ (IQR)	1.00 (3.10)	2.30 (2.30)	1.60 (2.30)	0.688	8.55 (5.70)	9.40 (7.05)	2.00 (2.60)	0.000000002 *	5.00 (1.50)	NA ^c^	1.60 (3.33)	0.008 *	2.20 (2.90)	2.10 (-)	2.20 (3.18)	0.954
Serum RAGE concentration ^a^ (IQR) (pg/mL)	1284.58 (6195.65)	754.34 (3635.61)	5579.31 (3414.10)	0.0003 *	3454.29 (44.02)	3469.16 (95.83)	3473.40 (3858.87)	0.824	3425.69 (853.46)	NA ^c^	3279.52 (1741.34)	0.836	NA ^b^	NA ^b^	NA ^b^	NA

HOMA2 IR: Insulin resistance; HOMA2%B: β-cell function; IQR: Interquartile range; NA: Non-applicable; OA: Orang Asli; RAGE: Receptor of advanced glycation end products. ^a^ Kruskal-Wallis test was applied. Assumptions were fulfilled. ^b^ We could not obtain comprehensive RAGE measurements for the OA. ^c^ We could not obtain sufficient sample size for prediabetic Indians. * significant if *p* < 0.05. In Malays, for corrected serum concentration of α-tocopherol, significant difference was found between pre-diabetes and non-diabetes (*p* = 0.002) as well as between pre-diabetes and diabetes (*p* = 0.031). In Malays, for HOMA2 IR, significant difference was found between non-diabetes and diabetes (*p* = 0.032). In Chinese, for HOMA2 IR, significant difference was found between pre-diabetes and diabetes (*p* < 0.0001). In Chinese, for HOMA2%B, significant difference was found between diabetes and non-diabetes (*p* < 0.0001) as well as between diabetes and pre-diabetes (*p* < 0.0001). In Indians, for HOMA2%B, significant difference was found between diabetes and non-diabetes (*p* = 0.006). In Malay, for serum RAGE concentration, significant difference was found between non-diabetes and diabetes (*p* = 0.006) as well as between pre-diabetes and diabetes (*p* = 0.003).

**Table 3 nutrients-12-03659-t003:** Ethnic comparison of median serum concentration of α-tocopherol (μmol/L), corrected serum concentration of α-tocopherol, HOMA2 IR, HOMA2%B and serum RAGE concentration (pg/mL), grouped in non-diabetes, pre-diabetes and diabetes.

	Non-Diabetes					Pre-Diabetes					Diabetes				
	Malay (*n* = 20)	Chinese (*n* = 20)	Indian (*n* = 17)	OA (*n* = 52)	*p* Value	Malay (*n* = 13)	Chinese (*n* = 21)	Indian	OA (*n* = 11)	*p* Value	Malay (*n* = 45)	Chinese (*n* = 35)	Indian (*n* = 29)	OA (*n* = 5)	*p* Value
Serum α-tocopherol concentration ^a^ (IQR) (μmol/L)	52.45 (38.31)	53.79 (45.76)	32.80 (20.40)	25.54 (16.52)	0.00003 *	41.03 (36.99)	50.21 (37.11)	NA ^c^	26.80 (9.26)	0.023 *	42.01 (42.55)	50.17 (39.01)	30.64 (29.30)	34.05 (17.90)	0.092
Corrected serum α-tocopherol concentration ^a^ (IQR)	0.9698 (0.2129)	0.9563 (0.2252)	0.9027 (0.4145)	0.9281 (0.4805)	0.416	0.6948 (0.1702)	0.8564 (0.2440)	NA ^c^	0.8938 (0.4498)	0.023 *	0.8542 (0.4077)	0.8596 (0.3601)	0.8496 (0.3837)	0.8911 (0.3764)	0.767
HOMA2 IR ^a^ (IQR)	0.0014 (0.0056)	0.0174 (0.0233)	0.0113 (0.0056)	0.0083 (0.0119)	0.003 *	0.0068 (0.0181)	0.0357 (0.0430)	NA ^c^	0.0111 (-)	0.005 *	0.0100 (0.0149)	0.0109 (0.0156)	0.0157 (0.0153)	0.0308 (0.0302)	0.141
HOMA2%B ^a^ (IQR)	1.00 (3.10)	8.55 (5.70)	5.00 (1.50)	2.20 (2.90)	0.00006 *	2.30 (2.30)	9.40 (7.05)	NA ^c^	2.10 (-)	0.0002 *	1.60 (2.30)	2.00 (2.60)	1.60 (3.33)	2.20 (3.18)	0.707
Serum RAGE concentration ^a^ (IQR) (pg/mL)	1284.58 (6195.65)	3454.29 (44.02)	3425.69 (853.46)	NA ^b^	0.092	754.34 (3635.61)	3469.16 (95.83)	NA ^c^	NA ^b^	0.301	5579.31 (3414.10)	3473.40 (3858.87)	3279.52 (1741.34)	NA ^b^	0.001 *

HOMA2 IR: Insulin resistance; HOMA2%B: β-cell function; IQR: Interquartile range; NA: Non-applicable; OA:Orang Asli; RAGE: Receptor of advanced glycation end products. ^a^ Kruskal-Wallis test was applied. Assumptions were fulfilled. ^b^ We could not obtain comprehensive RAGE measurements for the OA. ^c^ We could not obtain sufficient sample size for prediabetic Indians. * significant at *p* < 0.05. In non-diabetes, significant difference were found in serum α-tocopherol concentration between OA and Malay (*p* = 0.003) and between OA and Chinese (*p* < 0.0001), Indian and Chinese (*p* = 0.035). In pre-diabetes, significant difference was found in corrected serum α-tocopherol concentration between Malay and Chinese (*p* = 0.03). In non-diabetes, significant difference was found in HOMA2 IR between Malay and Chinese (*p* = 0.004). In pre-diabetes, significant difference was found in HOMA2 IR between Malay and Chinese (*p* = 0.006). In non-diabetes, significant difference were found in HOMA2%B between Malay and Chinese (*p* = 0.003), between OA and Indian (*p* = 0.046) and between OA and Chinese (*p* < 0.0001). In pre-diabetes, significant difference was found in HOMA2%B between Malay and Chinese (*p* < 0.0001). In diabetes, significant difference was found in serum RAGE concentration between Indian and Malay (*p* = 0.002).

**Table 4 nutrients-12-03659-t004:** The association factors of the different stages of development of type 2 diabetes with corrected serum α-tocopherol concentration, HOMA2 IR, HOMA2%B and serum RAGE concentration.

	Non-DM vs. Pre-DM		Pre-DM vs. DM		Non-DM vs. DM	
	*N* = 106	*n* = 46	*n* = 46	*n* = 109	*n* = 106	*n* = 109
	Crude OR (95% CI)	*p* Value	Crude OR (95% CI)	*p* Value	Crude OR (95% CI)	*p* Value
Low corrected serum α-tocopherol concentration ^b^	4.192 (2.005, 8.765)	0.0001 **	0.688 (0.333, 1.421)	0.312	2.885 (1.654, 5.031)	0.0002 **
High HOMA2 IR ^b^	2.423 (1.052, 5.582)	0.038 *	0.785 (0.358, 1.720)	0.546	1.903 (1.013, 3.573)	0.045 *
Low HOMA2%B ^b^	0.538 (0.219, 1.323)	0.177	5.657 (2.397, 13.349)	0.000076 **	3.046 (1.609, 5.766)	0.001 **
High serum RAGE concentration ^b^	0.885 (0.372, 2.107)	0.782	3.244 (1.454, 7.236)	0.004 *	2.870 (1.498, 5.496)	0.001 **

DM: Diabetes; HOMA2 IR: Insulin resistance; HOMA2%B: β-cell function; Non-DM: Non-diabetes; OR: Odds ratio; Pre-DM: Pre-diabetes; RAGE: Receptor of advanced glycation end products. abinary logistic regression were applied ^b^ Parameters are dichotomized with median as cut off point. * Significant at *p* < 0.05, ** Significant at *p* < 0.001.

**Table 5 nutrients-12-03659-t005:** Correlation of corrected serum α-tocopherol concentration with HOMA2 IR, HOMA2%B and serum RAGE Concentration.

	HOMA2 IR	HOMA2%B	Serum RAGE concentration
	r_s_ (*p* Value)	r_s_ (*p* Value)	r_s_ (*p* Value)
Corrected serum α-tocopherol concentration	−0.029 (0.699)	0.077 (0.308)	0.035 (0.633)

HOMA2 IR: Insulin resistance; HOMA2%B: β-cell function; RAGE: Receptor of advanced glycation end products. aSpearman’s correlation. Assumptions were fulfilled.

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
