# Peer review of "Vitamin E Levels in Ethnic Communities in Malaysia and Its Relation to Glucose Tolerance, Insulin Resistance and Advanced Glycation End Products: A Cross-Sectional Study"

_nutrients, 2020, doi:10.3390/nu12123659_

Round 1
Reviewer 1 Report
In the study Chua et al., explore the basis of ethnical variation in diabetes incidence in a Malaysian cohort comprising 4 distinct ethnic groups.
Following are the major concerns:
- Sub-stratification (based on ethnicity and diabetic status) leads to very small sample sizes.
- What were the exclusion/inclusion criteria?
- Was the data corrected with respect to diet/diabetes treatment/age/BMI? These are important confounders especially for RAGE levels.
- The variables mentioned in point 3 should be discussed.
Reviewer 2 Report
What is RAGE knockdown? Define it first, before discussing it.
identity rharked Eppendorf tubes? Do you mean marked Eppendorf tubes?
In your linear regression equation for determining alpha-tocopherol concentrations, what are B1 and B2, x1, x10 etc.?
I think you should remove the following sentence from your paper "The extrapolation of data could be debatable as it might be imprecise and thus, fails to capture the actual picture of the progression of beta-cell function decline in diabetes." It's as if you don't believe your own data i.e findings. Why don't you say that the findings of your study need to follow-up and verified by additional research studies.
Based on my understanding of the paper, I think you derived adjusted alpha-tocopherol concentrations of your subjects that took into account differences in the subjects' cholesterol and triglyceride concentrations. However, the "effective, true concentration" of alpha-tocopherol circulating in the blood is expressed by the amount of alpha-tocopherol per milliter of blood, irrespective of cholesterol or triglyceride concentrations.
Round 2
Reviewer 1 Report
All the concerns have been answered.